# Dynamic changes of enhancer and super enhancer landscape in degenerated nucleus pulposus cells

Guowang Li[1,2,*], Yuxiang Kang[2,*], Xiangling Feng[2,*], Guohua Wang[2,*], Yue Yuan[2], Zhenhua Li[1], Lilong Du[1], Baoshan Xu[1]

Inflammatory cascade and extracellular matrix remodeling have been identified as pivotal pathological factors in the progression of intervertebral disc degeneration (IDD), but the mechanisms underlying the aberrant activation of transcription during nucleus pulposus (NP) cell degeneration remain elusive. Super-enhancers (SEs) are large clusters of adjacent lone enhancers, which control expression modes of cellular fate and pathogenic genes. Here, we showed that SEs underwent tremendous remodeling during NP cell degeneration and that SE-related transcripts were most abundant in inflammatory cascade and extracellular matrix remodeling processes. Inhibition of cyclin-dependent kinase 7, a transcriptional kinase–mediated transcriptional initiation in trans-acting SE complex, constricted the transcription of inflammatory cascades, and extracellular matrix remodeling–related genes such as *IL1β* and *MMP3* in NP cells, meanwhile, also restrained the transcription of *Mmp16*, *Tnfrsf21*, and *Il11ra1* to retard IDD in rats. In summary, our findings clarify SEs control the transcription of genes associated with inflammatory cascade and extracellular matrix remodeling during NP cell degeneration and identify inhibition of the cyclin-dependent kinase 7, required for SE-mediated transcriptional activation, as a therapeutic option for IDD.

## Introduction

Low back pain (LBP) is a common disease, the economic loss and social burden caused by LBP in people aged 25–74 ranked fourth (Vlaeyen et al, 2018), of which 40% were related to intervertebral disc degeneration (IDD) (Zehra et al, 2022). The pathological process of IDD could be divided into three stages (Risbud & Shapiro, 2014): nucleus pulposus (NP) cells secreting pro-inflammatory cytokines are the earliest stage because of various factors (Phillips et al, 2013), such as genetics, aging, or abnormal load (Walter et al, 2011); second, pro-inflammatory factors and chemokines released by NP cells could promote degeneration of NP cells themselves and simultaneously recruit and activate immune cells such as macrophages and monocytes (Sun et al, 2020), and which could produce more inflammatory factors and chemokines to form inflammatory cascade (Kawaguchi et al, 2002); third, pro-inflammatory factors promote the expression of matrix metalloproteinase (MMP) leading to extracellular matrix (ECM) degradation (Millward-Sadler et al, 2009), subsequently painful nerve fibers and microvessels growing into intradiscal fissures, causing hyperalgesiainto and LBP (Ni et al, 2019), thus exploring the mechanism of inflammatory cascade from a new perspective is of great significance for retarding IDD.

Super-enhancers (SEs) are large clusters of multiple proximal enhancers (Li et al, 2021) and are enriched for high densities of transcription factors, cofactors, and epigenetic modifications (Pott & Lieb, 2015). It differs from lone enhancers (LEs) in sequence size, transcription factor binding density, ability to activate transcription, and sensitivity to transcription factor inhibitors (Mi et al, 2020). Current studies have found that SE-regulated transcription is dependent on bromodomain-containing protein 4 (BRD4) (Lee et al, 2017), the Mediator complex (Quevedo et al, 2019), the TF IIH complex containing cyclin-dependent kinase 7 (CDK7) (Glover-Cutter et al, 2009), and the transcription elongation complex (P-TEFb) containing CDK9 (Parua et al, 2020). CDK7 initiates transcription by phosphorylation of serine 5 (Ser5p) of Pol II C-terminal domain (CTD) (Kwiatkowski et al, 2014); CDK9 mainly phosphorylates serine 2 of Pol II CTD to promote transcriptionally paused Pol II to enter the transcription elongation stage, also known as Pol II release (Peterlin & Price, 2006). In addition, BRD4 promotes the assembly of super-enhancers by recruiting the Mediator complex and thus promote the release of Pol II from the paused state (Dawson et al, 2011), and CDK12/13 could accelerate the transcriptional elongation of Pol II (Liang et al, 2015). Therefore, it is generally believed that the key regulatory points of SE regulation of transcription, the Mediator complex, BRD4, and key CDKs are potential to be developed as new targets for the treatment of diseases (Ha Youn, 2018). However, the underlying mechanism of transcriptional abnormalities in intervertebral disc degeneration and whether CDK7 inhibitors can delay intervertebral disc degeneration remain unexplored.

[1]Department of Minimally Invasive Spine Surgery, Tianjin Hospital, Tianjin, China [2]Graduate School of Tianjin Medical University, Tianjin, China

Correspondence: baoshanxu99@tmu.edu.cn
*Guowang Li, Yuxiang Kang, Xiangling Feng, and Guohua Wang contributed equally to this work

In this study, we compared SEs between normal nucleus pulposus cells (NNP) and degenerated nucleus pulposus cells (DNP) and demonstrated that inflammatory factor leads to changes in enhancer and super-enhancer landscape, demonstrating that SEs are widely acquired at genes related to inflammatory factors and ECM-degrading enzymes in NP cells, such as *IL1β* and *MMP3*. Concomitantly, the function of CDK7, the phosphorylation of serine fifth in Pol II CTD, is also elevated in DNP. Transcriptomic data in DNP cells pretreated with the 50 nM CDK7-specific inhibitor THZ1 (TDNP) showed that the transcription of *IL1β*, *CSF2*, *TNFRSF1B*, and *MMP3* was repressed. Finally, we found that IDD rats were more sensitive to the inhibition of CDK7 than normal, and that this inhibition also curbed the genes related to inflammatory cascade and ECM remodeling. This study provides the first characterization of enhancer and super-enhancer landscape before and after the degeneration of NP cells and important insights into retarding IDD progression.

# Results

### Inflammatory factors change enhancer repertoire of NP cells in vitro

To explicit the influence of inflammatory factors on the enhancer and super-enhancer landscape of NP cells, we performed chromatin immunoprecipitation sequencing (ChIP-seq) analysis of H3K27 acetylation (H3K27ac) to NNP and DNP cells. H3K27ac sites were used to seek out active enhancers through narrow peak calling. Composite plots of normalized ChIP-seq signals for H3K27ac in NNP and DNP samples indicated that DNP had a higher signal intensity for H3K27ac overall than NNP (Fig 1A). We identified 51,212 H3K27ac binding markers commonly existed in NNP and DNP samples, besides 10,423 and 12,585 sites that were specifically bound in NNP and DNP, respectively (Fig 1B). For subsequent analysis, we entitled three sets of binding sites: common (existed in NNP and DNP samples); NNP-specific (specificly existed in NNP sample); and DNP-specific (specificly existed in DNP sample). Composite maps and heatmaps of the three sets of binding sites in NNP and DNP demonstrate that inflammatory factors might lead to a dramatic shift in the enhancer and super-enhancer landscape of NP cells (Fig 1C–E).

### DNP-specific enhancers regulate inflammatory factor–induced gene expression

Bedtools of R were used to assign enhancer elements to genes. The number of genes to which enhancers were assigned in the three groups showed the DNP-specific enhancers were overwhelmingly assigned to one gene, whereas more than half of the NNP-specific enhancers were not even assigned to one gene. The DNP-specific group and NNP-specific group had similar distance ratios to nearby genes (Fig 2A). KEGG enrichment analysis of genes proximal to DNP-specific enhancers revealed that the focal adhesion might play an important role in intervertebral disc degeneration (Chen et al, 2022) (Fig 2B). Using sample-matched RNA-Seq gene expression data, we determined the functional relevance of identified enhancer

elements. When comparing DNP with NNP, RNA-seq data showed 304 significantly up-regulated genes and 203 significantly down-regulated genes, with q value < 0.01 and |log$_2$fold change| ≥ 1 (Fig 2C and D). Overall, the expression of genes which were assigned to by DNP-specific enhancers was significantly increased compared with genes associated with common enhancers and NNP-specific enhancers. The expression changes of NNP-specific enhancer–related genes showed no statistical significance when compared with common enhancer-related genes (Fig 2E). These data suggest that inflammatory factors can alter the enhancer landscape and that the higher the H3K27ac signal level of the enhancer, the higher the expression level of its related genes.

### Changes and roles of super-enhancers in NP cells during degeneration

Aberrant activation of transcriptional programs must be involved in the pathogenesis of IDD. To look into the underlying impacts of altered SEs in IDD progression, enhancer activity was positively correlated with the extent of H3K27ac modifications bound to its DNA sequence, SEs have a high density of H3K27ac. We stitched component enhancers that occurred within 12.5 kb and ordered them according to their H3K27ac signal using the ranking of super-enhancers algorithm. Stitched enhancers that occur above the inflection point of H3K27ac signaling are SEs, and enhancers below the inflection point are considered LEs. We identified 248 DNP-specific SEs, 195 NNP-specific SEs, and 1,022 common SEs, which were present in both NNP and DNP (Fig 3A). These data again suggest that the H3K27ac-binding site of the SEs undergoes extensive changes during degeneration. DNP-specific SEs showed that the level of H3K27ac modification increased at their distinct enhancer sites. In contrast, NNP-specific SEs exhibited reduced enrichment of H3K27ac at their enhancer sites. The average distribution of H3K27ac was enriched across all identified SEs compared with LEs (Fig 3B). The integration of SEs and LEs data with matched RNA-Seq data showed that the DNP-specific and common SEs-related genes were higher expressed than LEs-related genes, whereas the expression changes of NNP-specific SEs-related genes were not different from LEs-related genes (Fig 3C). Furthermore, the fold change in expression of genes associated with DNP-specific SEs was significantly higher compared with genes associated with common or NNP-specific SEs (Fig 3D). KEGG enrichment analysis illustrated genes associated with DNP-specific SEs were enriched in TNF-α signaling pathway during NP cells degeneration (Fig 3E). These results suggest that the SEs landscape is altered in NP cells degeneration and DNP-specific SEs contributes greatly to transcriptional abnormalities.

### Transcription factor binding predicted by motif analysis

To predict the candidate transcription factors (TFs) that regulate the enhancer landscape of NP cells. We performed motif discovery analysis. The first 10 transcription factor motifs defined by *P*-values in the common, NNP-specific, and DNP-specific enhancer groups are shown in Fig 4A, and *P*-values reflect the abundance of TF motifs in enhancer sequences compared with background sequences. Fig 4B shows the percentage of enhancer sequences with potential TF motifs.

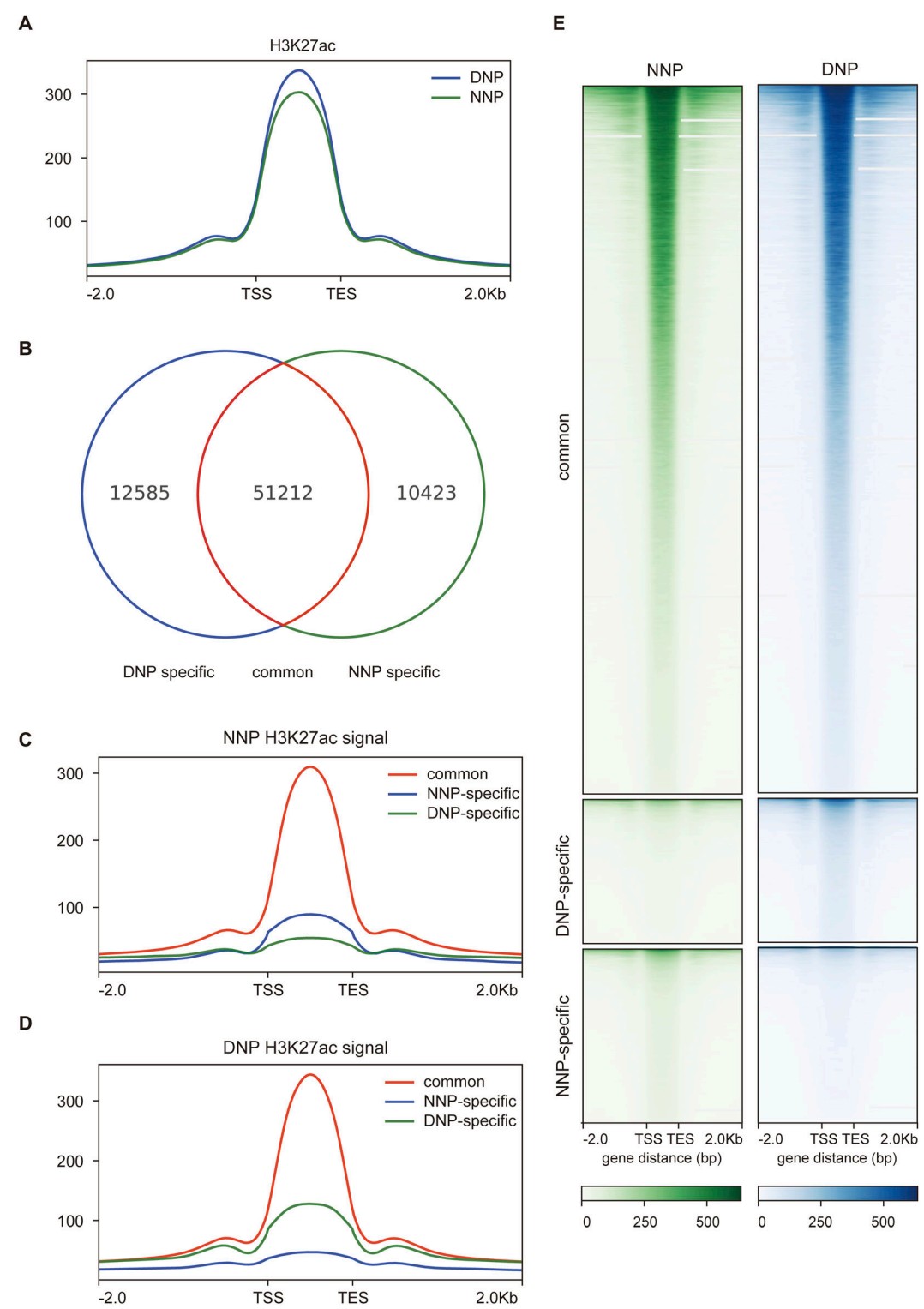

**Figure 1.   H3K27ac ChIP-seq signal characterization in NNP and DNP cells.**
**(A)** Composite plots of normalized ChIP-seq signals for H3K27ac in NNP and DNP cells. **(B)** Narrow peak calling was applied on H3K27ac NNP and DNP samples. 51,212 peaks overlapped between them; 10,423 and 12,585 were only identified in NNP or DNP samples, respectively. **(C)** Composite plot showing normalized ChIP-seq signals of H3K27ac for three groups including common, NNP-specific, and DNP-specific peaks using NNP files. **(D)** Composite plot showing normalized ChIP-seq signals of H3K27ac for three groups including common, NNP-specific, and DNP-specific peaks using DNP files. The x-axis represents distance from the H3K27ac peaks and y-axis represents probability density. **(E)** The heatmap of normalized H3K27ac ChIP-seq signal in NNP and DNP cells. The rows show 2 kb around the H3K27ac peak centre.

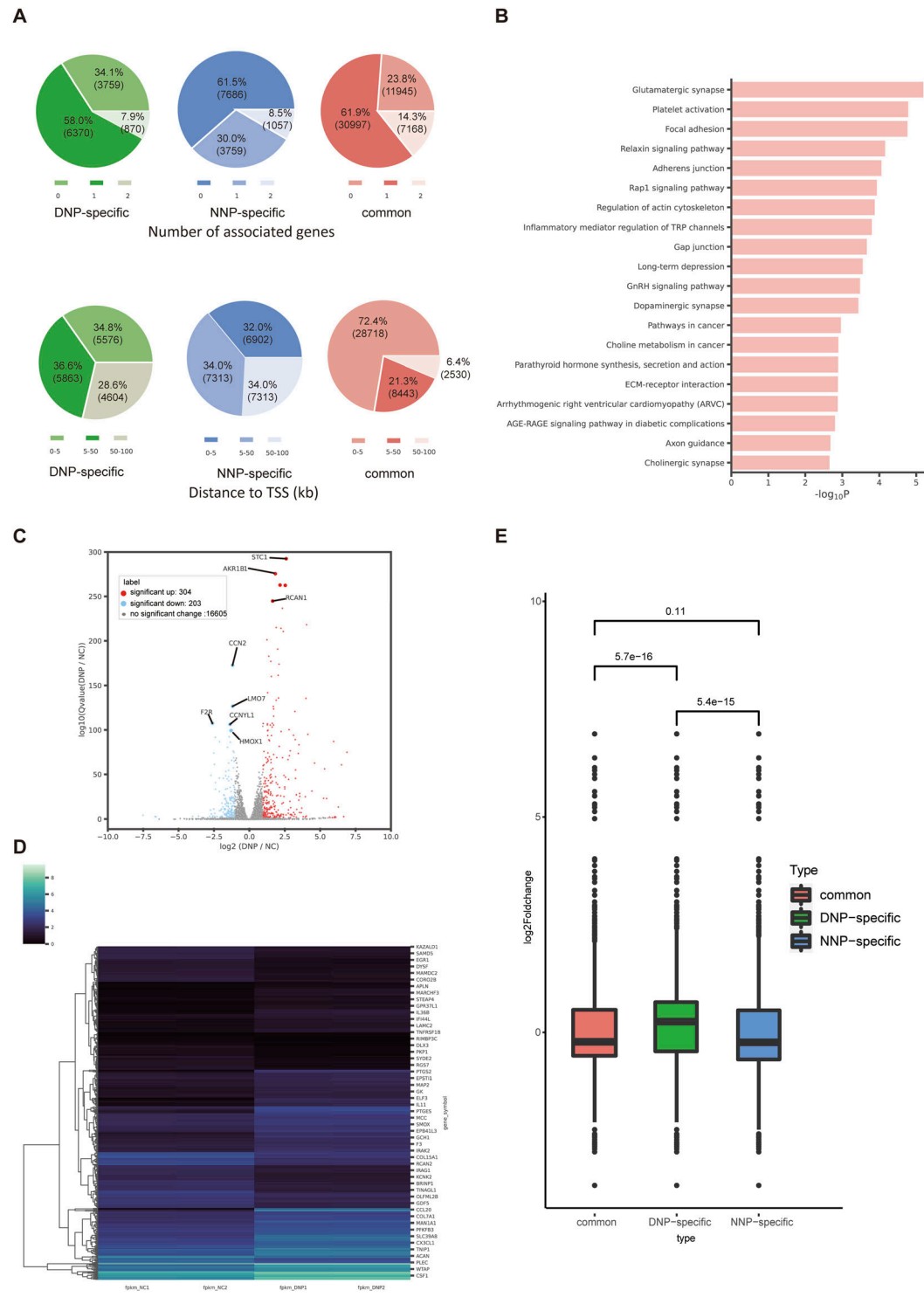

**Figure 2. Enhancers—gene annotation.**
**(A)** Bedtools were used to annotate enhancers to genes. Pie chart showing how many genes are annotated as enhancer elements. Pie chart illustrating the distance between enhancers and their annotated genes (kb to transcription start site). **(B)** KEGG enrichment analysis of genes proximal to the DNP-specific enhancers. **(C)** Volcano plot showing differentially expressed genes (DNP versus NNP, n = 507). **(D)** Heatmap of gene expression values in NNP and DNP cells. Rep1 and Rep2 represent two biological replicates. **(E)** Comparison between genes, respectively, associated with three groups including common, NNP-specific, and DNP-specific enhancers.

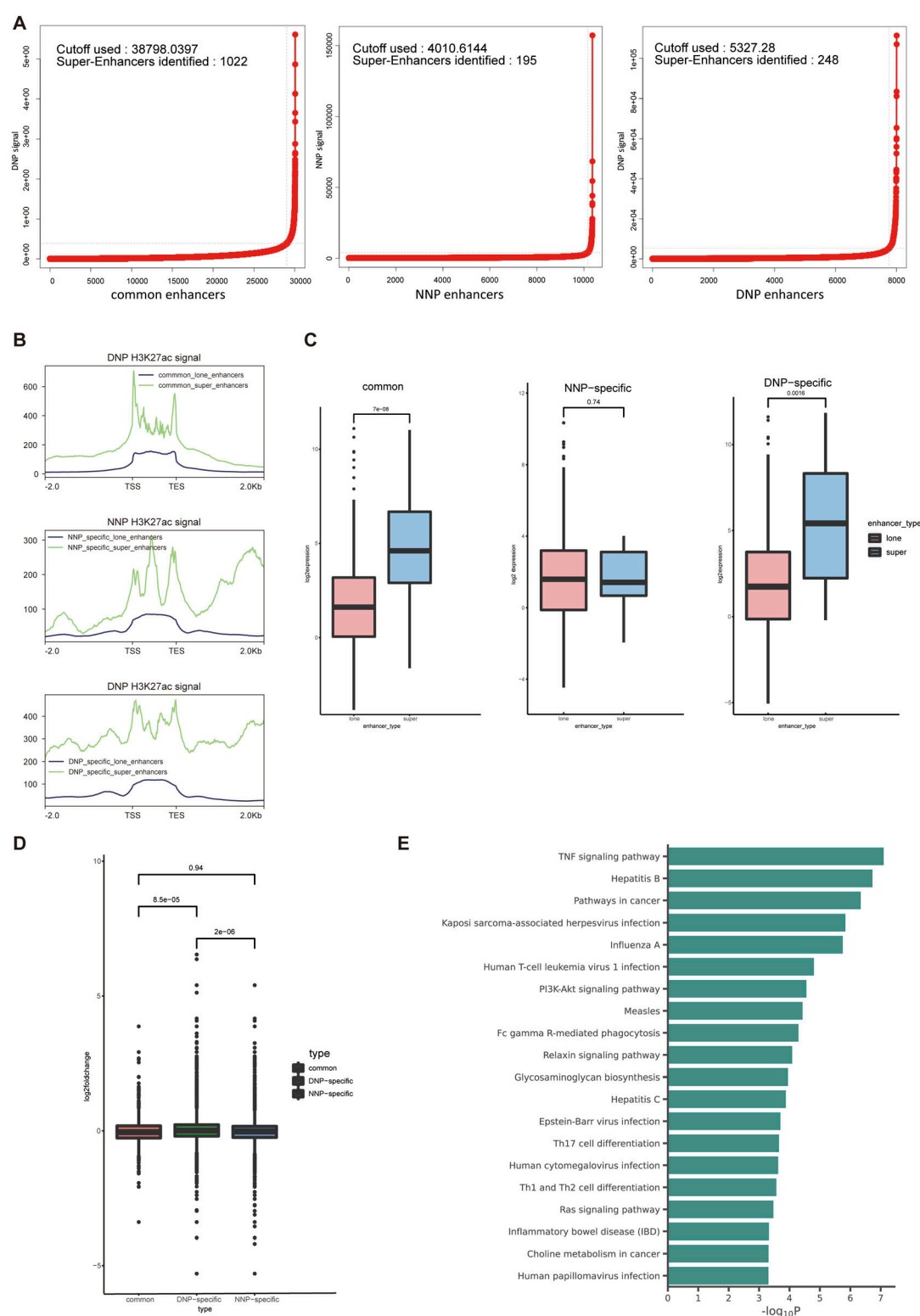

**Figure 3.   Characterization of the SE landscape of NNP and DNP cells and their annotation to genes.**
**(A)** Super-enhancer profiles and enhancer rank of all three H3K27ac peak categories defined by H3K27ac ChIP-seq signals. Cutoffs for distinguishing between lone enhancers (LEs) and super-enhancers (SEs) are shown as dashed lines. **(B)** Composite plot of normalized ChIP-seq signals of H3K27ac for LEs and SEs identified by all three H3K27ac peak categories. **(C)** Box plots depict the differences in gene expression between LEs-related genes and SEs-related genes within each of the three groups. **(D)** Box plots showing the relative expression levels of common SE-related genes, NNP-specific SEs-related genes, and DNP-specific SEs-related genes. **(E)** KEGG enrichment analysis of genes assigned to DNP-specific SEs.

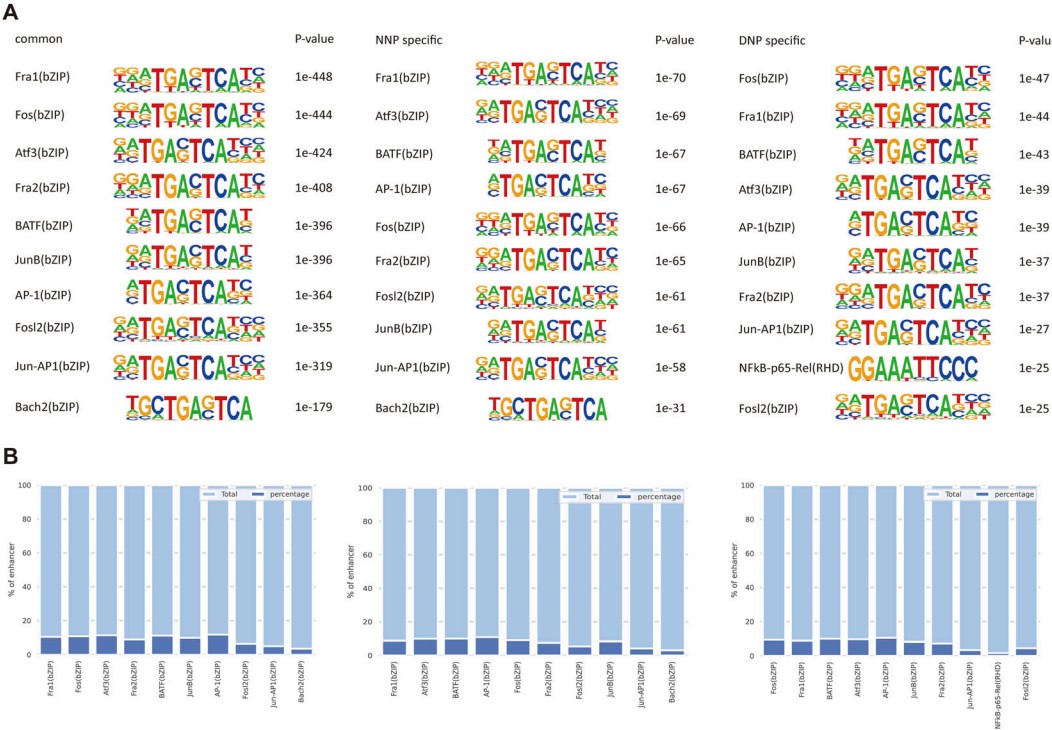

**Figure 4. Motif analysis of enhancer elements.**
**(A)** The top 10 identified transcription factor motifs in each of the three categories, common, NNP-specific, DNP-specific ranked by *P*-value. **(B)** The bar plots show for each category the percentage of enhancers having the specific motif.

## Inhibition of CDK7 activity attenuates DNP cells degeneration

To deliberate the functions of CDK7 during NP cells degeneration, we inhibited CDK7 in NP cells using THZ1, which is a specific inhibitor of CDK7. THZ1 can covalently bind to cysteine 312 of CDK7 to inhibit the kinase activity of CDK7, thereby inhibiting the phosphorylation of CDK7 on the fifth serine of Pol II CTD. First, we examined the effect of THZ1 on NP cell viability, CCK-8 results of NNP cells treated with different concentrations of THZ1 for 24 h showed that 50 nM of THZ1 was the largest concentration that did not affect cell viability (Fig 5A). We further examined the cytoprotective effect of THZ1 using live and dead cell staining. Correspondingly, 50 nM THZ1 treatment reduced the proportion of dead cells, whereas higher concentrations of THZ1 promoted apoptosis (Fig 5B), so we chose 50 nM as the fixed intervention, and the pretreatment of DNP with THZ1 for 2 h was named as TDNP.

As expected, the expression level of Ser5p in Pol II CTD was higher in DNP than in NNP, whereas it was down-regulated in TDNP, correspondingly, MMP13 showed the same trend (Fig 5C and E). Then, toluidine blue staining also demonstrated that THZ1 had a rescue effect on the breakdown of ECM during NP cell degeneration, and SA-β-Gal staining demonstrated that THZ1 could partially attenuate the senescence of DNP cells (Fig 5D).

## Analysis of SE-driven target genes rescued by THZ1 in NP cells

Subsequently, we intently unveiled to the molecular mechanism by which THZ1 retards the degeneration of NP cells. To this end, we combined DNP-specific SE drive genes with transcriptomic data from three sets of NP cells. CDK7 plays a key role in trans-SE complex driven transcription, and the related transcripts promoted by SE are particularly sensitive to CDK7 inhibition compared with LEs. Hence, our analysis concentrated on genes down-regulated by THZ1. As shown in Fig 6A, we overlapped DNP-specific SE drive genes with THZ1 rescue genes, which were up-regulated in DNP cells and reversed by THZ1 treatment, and found that THZ1 treatment can reduce the expression of these 81 up-regulated genes, which was originally driven by DNP-specific SEs. KEGG enrichment analysis revealed these 81 genes were also enriched in the TNF-α signaling pathway (Fig 6B), as shown in Fig 6C, including *IL1β*, *CSF2*, *TNFRSF1B*, and *MMP3*, and whose H3K27ac ChIP-seq and RNA-seq profiles were pictured in Fig 6D. Notably, these potential THZ1-rescued genes were identified, including *IL1β*, *CSF2*, *TNFRSF1B*, and *MMP3*, we performed qRT-PCR on NNP, DNP, and TDNP groups cultured in vitro, it was verified that these transcript level was increased in DNP, whereas these transcript level was decreased in the TDNP group (Fig 6E).

## THZ1 ameliorates the progression of IDD in vivo

To simulate IDD caused by rupture of the annulus fibrosus, we established rat IDD models using acupuncture. Over the next 2 wk, THZ1 was injected intraperitoneally for 5 consecutive days, followed by a 2-d rest. At 2 wk after puncture, magnetic resonance imaging (MRI) was used to assess the degree of IDD in rats. As expected, the modeling of IDD group was successful. However, after THZ1 treatment, MRI images showed a significant increase in signal

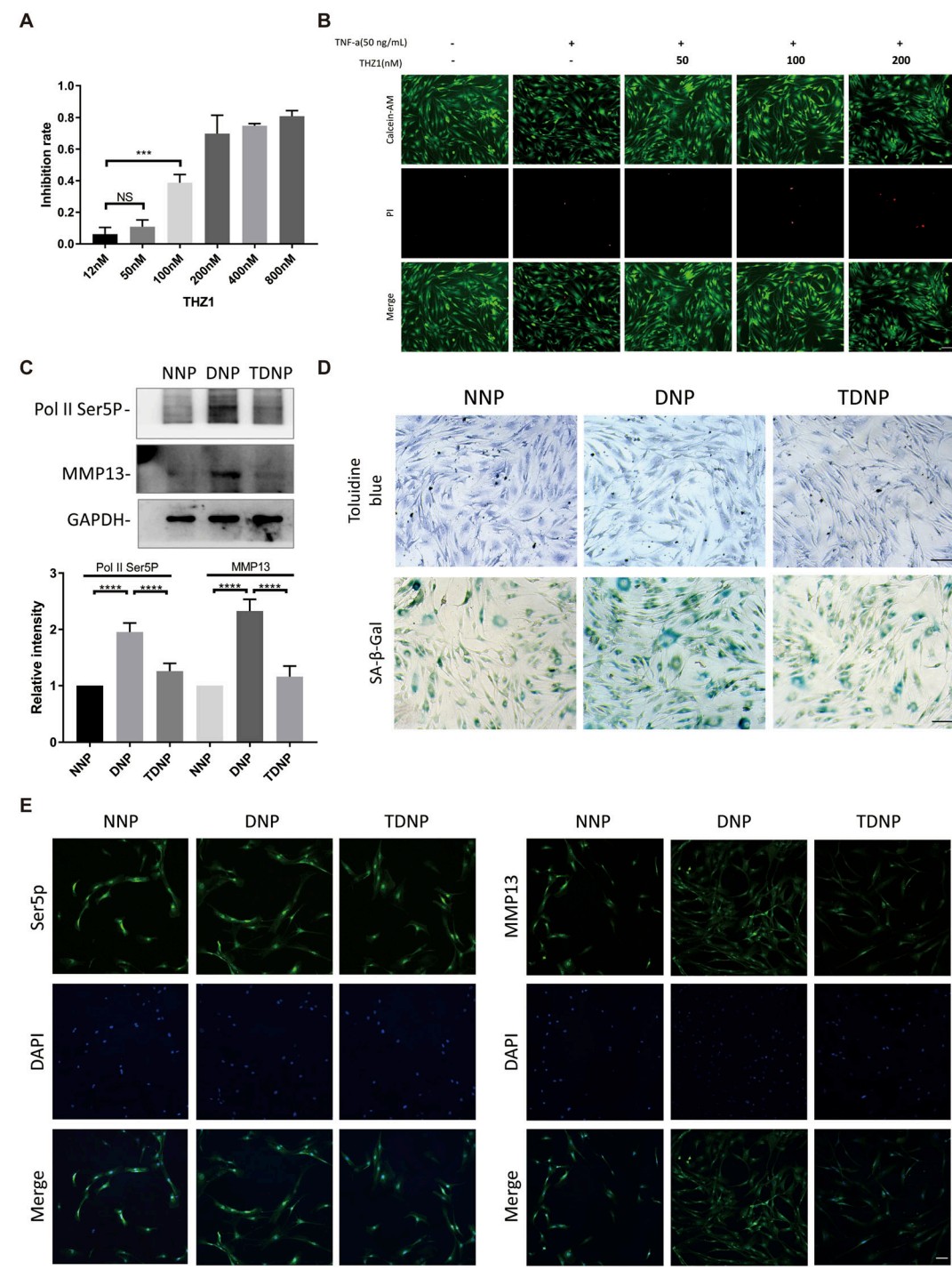

**Figure 5.  THZ1 inhibits degradation of human nucleus pulposus (NP) cells.**
**(A)** Cytotoxic effects of THZ1 on human NP cells were assessed using cell counting kit-8 cell viability and toxicity assays at 24 h (n = 3 samples for each group). **(B)** Representative fluorescence photomicrograph of live and dead cell staining of NP cells treated with different concentrations THZ1 for 24 h. **(C)** Western blot analysis of Pol II Ser5p and MMP13 levels in NNP, DNP, and TDNP cells. Quantitative analysis was shown on the underside of WB results. **(D)** Toluidine blue staining and SA-β-Gal staining of NNP, DNP, and TDNP cells (n = 3 samples for each group). **(E)** IF analysis of Pol II Ser5p and MMP13 in NNP, DNP, and TDNP cells. Scale bars, 100 $\mu$m (B, D, E). Data information: in (A, C), data are presented as mean ± SEM. NS, not significant, *$P$ < 0.05, **$P$ < 0.01, ***$P$ < 0.001, ****$P$ < 0.0001 ($t$ test).

intensity (Fig 7A), furthermore Alcian blue, toluidine blue, and Sirius red staining were performed (Fig 7B). Matrix remodeling events driven by replacement of hydrophilic proteoglycans and non-collagen with fibrillar collagen matrices are characteristic of degenerative disc disease; therefore, we assessed the structure and composition of the ECM of the intervertebral disc.

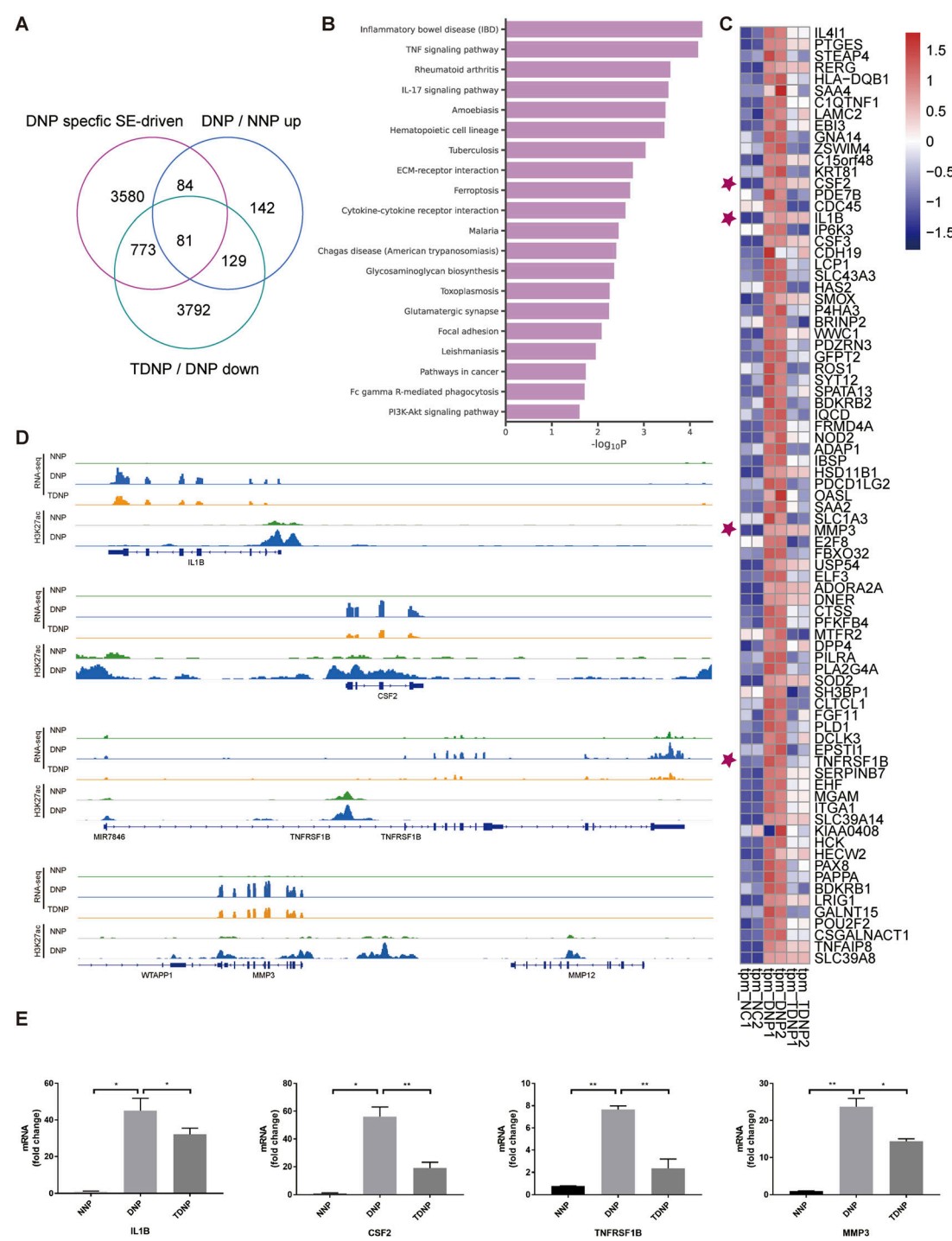

**Figure 6.  Identification of DNP-specific SE-driven and THZ1-rescued target genes in NP cells.**
**(A)** Venn plot of DNP-specific SE-driven genes, up-regulated genes in DNP versus NNP, and down-regulated genes in TDNP versus DNP. **(B)** KEGG enrichment analysis of the 81 DNP-specific SE-driven and THZ1-rescued target genes. **(C)** Heatmap of the 81 DNP-specific SE-driven and THZ1-rescued target genes expression values in NNP, DNP, and TDNP cells. Rep1 and Rep2 represent two biological replicates. **(D)** Gene tracks showing representative H3K27ac ChIP-seq profiles of *IL1β*, *CSF2*, *TNFRSF1B*, and *MMP3* genes in NNP, DNP cells, and the corresponding RNA-seq profiles in NNP, DNP, and TDNP cells. **(E)** qRT-PCR analyses were performed for *IL1β*, *CSF2*, *TNFRSF1B*, and *MMP3* detection in NNP, DNP, and TDNP cells (n = 3 samples for each group). Data information: in (E), data are presented as mean ± SEM. NS, not significant, *P < 0.05, **P < 0.01, ***P < 0.001 (t test).

Total collagen content and collagen fiber orientation were visualized by Sirius red staining and polarized light microscopy (Fig 7C), consistent with the degenerative changes observed with previous tissue staining, IDD discs showed disorganized fibrillar collagen in the NP compartment, and fibrosis and ECM remodeling was attenuated by THZ1 treatment in the NP region. Taken together, THZ1 treatment partially attenuated ECM remodeling in IDD.

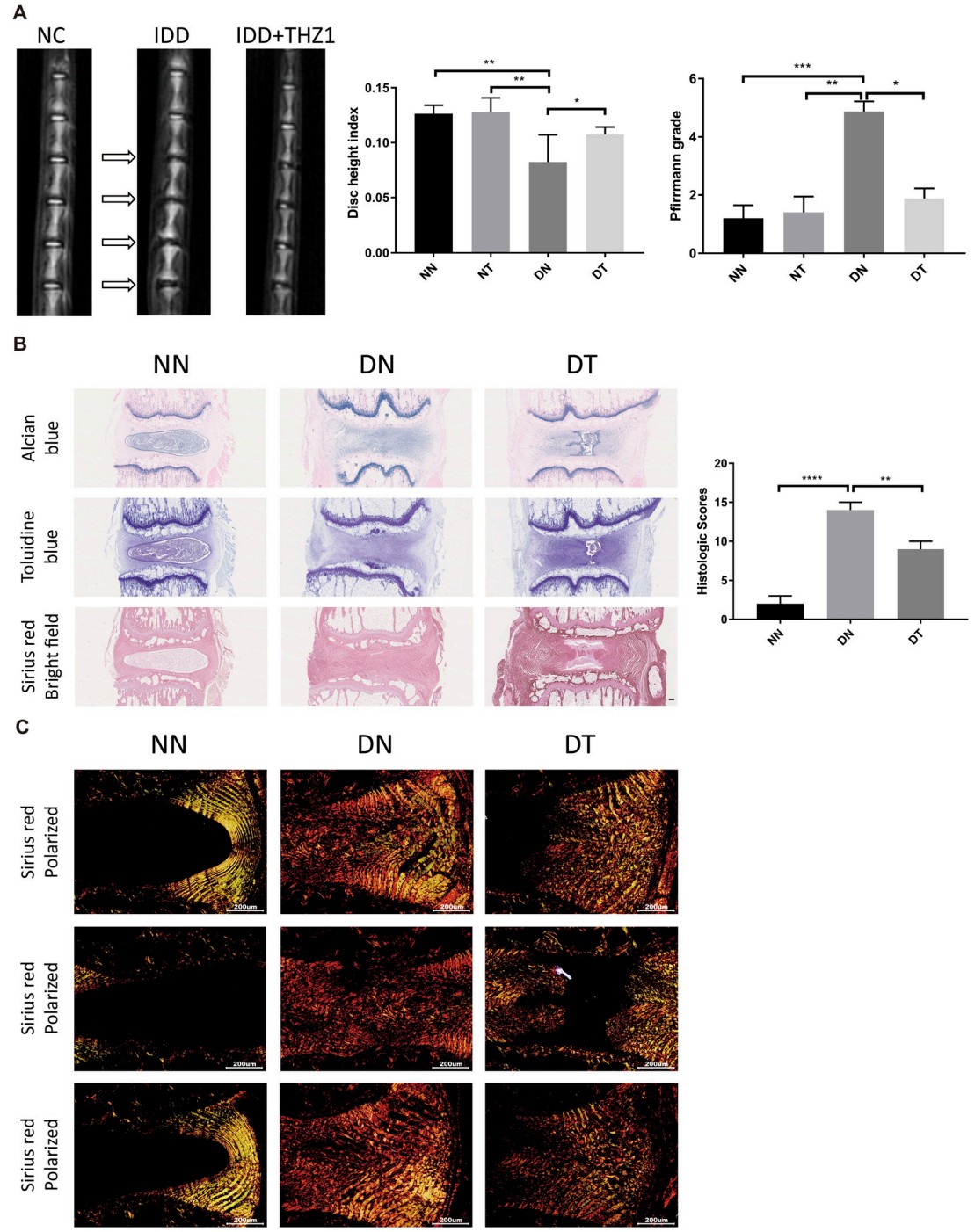

**Figure 7. Role of THZ1 in rat intervertebral disk degeneration models.**
**(A)** Magnetic resonance imaging images, disk height index, and Pfirmann grade of the rat models with and without THZ1 treatment at 2 wk after operation (n = 3 samples for each group). **(B)** Alcian blue staining, toluidine blue staining, and Sirius red with bright field of different types of disc sections in rat models, the histological grades evaluated in NN, DN, and DT groups are shown on the right (n = 3 samples for each group). **(C)** Collagen content in AF and NP tissue visualized by Sirius red with polarized microscopy. Scale bars, 200 $\mu$m (B, C). Data information: in (A, B), data are presented as mean ± SEM. NS, not significant, *P < 0.05, **P < 0.01, ***P < 0.001 (t test).

### THZ1 regulates differentially expressed genes associated with inflammatory cascades and ECM remodeling in rat IDD models

We performed RNA-seq analysis on the intervertebral disc tissue of IDD rat models and divided them into normal group (NN), normal and THZ1 treatment group (NT), only degeneration group (DN), and degeneration and THZ1 treatment group (DT). Only 23 (8 down-regulated and 15 up-regulated) genes were differentially expressed in the NT group compared with the NN group. However, compared with the DN group, the DT group had 650 up-regulated genes and

499 down-regulated genes with q value < 0.01 and |log₂fold change| ≥ 1 (Fig 8A). The Venn and heatmaps of the genes rescued by THZ1 are shown in Fig 8B and C, and the KEGG map of the genes rescued by THZ1 is shown in Fig 8D, providing further evidence that THZ1 could attenuate ECM remodeling in rat models of IDD. Changes in inflammation-related THZ1-rescued genes in rat IDD models are shown in Fig 8E, and those associated with ECM remodeling are shown in Fig 8F.

# Discussion

In the related studies of renal ischemia-reperfusion injury (IRI) (Wilflingseder et al, 2020), the binding sites of BRD4 and H3K27ac changed after IRI and were related to gene expression. For example, the BRD4 and H3K27ac signals at the enhancer of *Slc34a1* and *Kl* were significantly decreased, which was consistent with the significantly decreased expression of *Slc34a1* and *Kl* genes after ischemic injury. On the contrary, more BRD4 and H3K27ac were enriched in the enhancer and promoter of the up-regulated *Havcr1* gene after injury. *Havcr1* is expressed in the damaged proximal tubule cells and is involved in clearing apoptotic cells and cell fragments in the lumen of the tubule (Ichimura et al, 2004). In addition, the application of BRD4 inhibitor JQ1 after IRI resulted in impaired kidney recovery and increased mortality in mice. In the study of comparing the SE landscape between normal liver and hepatocellular carcinoma cells, it was shown that the SE landscape of cis-acting was extensively reprogrammed in the process of hepatocarcinogenesis, in which HCC cells obtained SEs at the oncogenes including *sphingosine kinase 1 (SPHK1)* to drive their vigorous expression (Tsang et al, 2019). Complex interactions between transcription factors (TFs), histone modifications, and cofactors for genomic regulatory elements such as SE mediate dramatic reprogramming of the gene expression reprogram in cell identity and development (Whyte et al, 2013). In this study, we have demonstrated the cis-regulatory SE landscape changes dramatically during human NP cell degeneration and overall level of H3K27ac increased during degeneration. Furthermore, among the reported SE-related transcriptional cofactors, we chose to delve deeper into the role of CDK7; first, we found that the function of CDK7 was up-regulated in human DNP cells, namely, the serine fifth phosphorylation in the CTD of Pol II. TNF-α and IL1β are known to be the main pro-inflammatory cytokines involved in the pathophysiology of IDD (Le Maitre et al, 2007), and external stimuli such as abnormal stress can promote the expression of IL1β and TNF-α in NP tissues (Lyu et al, 2021). NP cells are known to be an important source of inflammatory factors, in addition, the paracrined TNF-α and IL1β of NP cells can form a positive feedback loop through the NF-κB signaling pathway, and the TNF-α signaling pathway can stimulate the expression of *CSF2* and chemokines, including *CCL3*, *CCL4*, and *C-X-C motif chemokine 10* (*CXCL10*) (Mundra et al, 2016), which play an important role in leukocyte recruitment and leukocyte activation (Arcuri et al, 2009), and simultaneously promotes the expression of *MMP3* to remodel the ECM (Séguin et al, 2005). Therefore, breaking the positive feedback loop that forms the inflammatory cascade from a new perspective is urgently needed to attenuate the progression of IDD.

In the present study, we observed the association of SEs with multiple key inflammation–related and ECM remodeling–related genes in DNP cells; these SEs were markedly absent in NNP, suggesting that they were acquired during degeneration. For example, we observed acquisition of SE at multiple genes in DNP cells, including *IL1β*, *MMP3*, and *TNFRSF1B*. Dysregulation of H3K27ac signals at the SE may have profound effects on the global gene expression program in DNP cells. Regarding inflammation-related genes, we observed that H3K27ac signaling was enriched at SEs of *IL1β*, *CSF2*, and *TNFRSF1B*, and regarding genes associated with ECM remodeling, we also observed increased SEs of *MMP3* in DNP cells, and perturbation of the SE machinery has been proven to effectively inhibit the SE-related genes obtained in DNP cells, and the transcription levels of the above key genes are reduced in TDNP cells, the up-regulation of *MMP13* expression, NP cells apoptosis, and senescence were eventually reversed by THZ1, and because combined Chip-seq and RNA-seq analysis showed that THZ1 did not directly regulate these phenotypes, which may be secondary consequences of the expressional reversal of these DNP-specific SE-driven inflammatory genes. At the same time, intraperitoneal injection of THZ1 in rats can significantly delay the degradation of ECM in intervertebral disc degeneration. Interestingly, the transcriptome changes in NT/NN were significantly weaker than those in DT/DN, suggesting degenerated discs were more sensitive to THZ1 treatment, suggesting that SEs might play an important role in the disease process in IDD rats. This observation demonstrates the feasibility of intervention by CDK7-specific inhibitor THZ1.

In single-cell transcriptome (scRNA-seq) analysis of different human NP tissues, the proportion of *COL6A3* and *COL15A1* expression increased in fibrochondrocytes during NP cell degeneration (Li et al, 2022). The same results were seen in DN samples from rats, and THZ1 reduced fibrosis-related genes including *Col15a1*, *Col18a1*, and *Col14a1*. It has been reported that if CCL2 activity is blocked, the infiltration of macrophages into the intervertebral disc and the associated inflammatory and pain responses may be limited (Nakawaki et al, 2020), and the expression of *TNFRSF1A* and *TNFRSF1B* is increased in herniated NP tissues and showed that TNF-α and TNFRSF1B protein levels were positively correlated with pain (Bachmeier et al, 2007). In rat RNA-seq data, THZ1 treatment reduced the elevated expression of *Ccl2*, *Cxcl14*, *Tnfrsf21*, *Il11ra1*, and *Cx3cl1* in DN samples, which may contribute to leukocyte migration and inflammatory cascades.

Currently, studies have shown that TFs are often combined in the enhancer and super-enhancer region, and which can coordinate the symphony of its own gene and other core TF genes (Novershtern et al, 2011), resulting in interconnected autoregulation circuit (Neph et al, 2012). However, the exact mechanism is unclear; therefore, our analysis of motif discovery in enhancers reveals which potential transcription factors mediate the transcription initiation of NP degeneration and the maintenance of degeneration-associated gene expression programs. The heterodimeric transcription factor AP-1 composed of members of the c-Fos and c-Jun families of transcription factors, the transcription of inflammatory cytokines, and MMPs is regulated by the transcription factor c-Fos/activator protein-1 (AP-1), and the selective c-Fos/AP-1 inhibitor T-5224 can delay degeneration and attenuate pain in rat models of

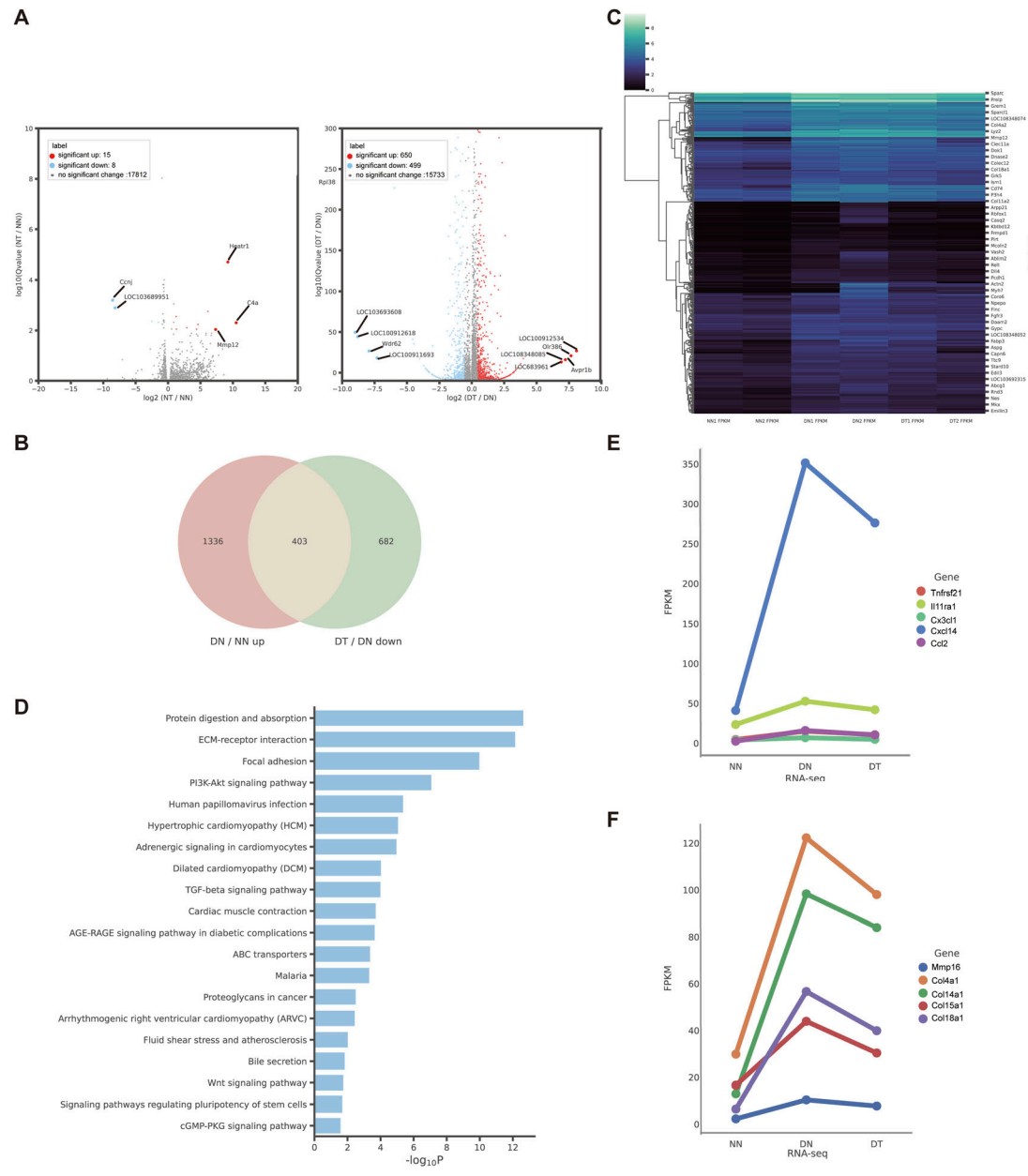

**Figure 8. Transcriptional consequences of CDK7 inhibition in the rat IDD models.**
**(A)** Volcano plot showing 23 differentially expressed genes (NT versus NN, n = 2 samples for each group), volcano plot showing 1,149 differentially expressed genes (DT versus DN, n = 2 samples for each group). **(B)** Venn plot of up-regulated genes in DN versus NN and down-regulated genes in DT versus DN. **(C)** Heatmap of the 403 THZ1-rescued target genes expression values in NN, DN, and DT samples of rat IDD models. Rep1 and Rep2 represent two biological replicates. **(D)** KEGG enrichment analysis of 403 THZ1-rescued target genes in rat IDD models. **(E)** Expression changes of inflammation-related genes in NN, DN, and DT samples of rat IDD models. **(F)** Expression changes of extracellular matrix remodeling–related genes in NN, DN, and DT samples of rat IDD models.

IDD (Makino et al, 2017), the underlying mechanism may be the landscape changes of SEs in DNP are mediated by c-Fos and AP-1.

In conclusion, the current study demonstrates that targeting transcriptional activation by inhibiting CDK7 is a promising IDD therapeutic strategy. Furthermore, this work provides important insights into the molecular pathogenesis of IDD by characterizing the cellular SE landscape and transcriptional profile of human NP cells, and that of rat IDD. Further translational studies of SE complex inhibitors such as THZ1 are anticipated to benefit patients with IDD in the future.

# Materials and Methods

## Isolation and culture of nucleus pulposus cells

NP tissue samples were dissociated from surgically obtained intervertebral discs as much as possible, cut into small pieces, and treated with 150 U/ml collagenase type II (A004174; SangonBiotech) for 2 h at 37°C. Digests were centrifuged at 225*g*, then resuspended in Dulbecco's Modified Eagle's Medium/F-12 (DMEM/F-12; Gibco)

containing 10% FBS (Invitrogen) and 1% penicillin–streptomycin (Sigma-Aldrich) at 37°C, 5% CO2. When grown to confluence, cells were digested with 0.25% trypsin–EDTA (25200056; Gibco) and expanded. NP cells from passage 2 to passage 5 were seeded into experimental plates for subsequent experiments. NP cells stimulated with 50 ng/ml TNF-α (300-01A; Peprotech) for 24 h were used as DNP. Unstimulated cells served as relatively NNP. DNP cells pretreated with CDK7 inhibitor THZ1 (A14192; Adooq) were named TDNP. For in vitro experiments, THZ1 was dissolved in DMSO to the concentration of 5 mM as a stock solution.

## Animal studies

All nursing and experimental protocols for rats were approved by the Ethics Committee of Tianjin Hospital. A total of 48 male Sprague–Dawley rats, aged 8 wk, were purchased from the Tianjin Orthopaedic Research Institute. As with the previous mature methods, 90 mg/kg ketamine and 10 mg/kg methylthiazide were injected intraperitoneally and placed in a prone position. The skin was sterilized with ethanol, and 21 G needles were used to puncture from the back of the intervertebral disc, rotated 180°, and maintained for 10 s. The remaining 12 rats did not receive surgical intervention as a negative control. After surgery, wound care was provided to prevent infection. It was dissolved in 5% DMSO + 45% PEG 300 + ddH$_2$O according to the instructions for in vivo use of THZ1 (A14192; Adooq). IDD rats were treated with continuous intraperitoneal injections for 5 d, followed by 2 d of drug withdrawal, for two cycles (14 d in total). The frequency of subsequent injections was once a week until 12 wk after surgery. Each rat was injected with approximately 0.83 mg of THZ1 per injection. All rats were divided into normal and non-THZ1–treated group (NN), normal and THZ1-treated group (NT), degeneration without THZ1-treated group (DN), and post-degeneration THZ1-treated group (DT).

## ChIP-seq and data analysis

According to the mature method, the fixation and termination of cross-linking of NNP and DNP cells at room temperature were treated with 1% formaldehyde for 10 min and 125 mM glycine for 5 min, respectively. After cells were washed with PBS, cells were scraped in ChIP lysis buffer (1% SDS, 10 mM EDTA, and 50 mM Tris–HCl, pH 8.0). DNA fragments with the size range of 100–500 base pairs were obtained using the Bioruptor sonicator. 2 $\mu$m g H3K27ac antibody (ab4729; Abcam) was used to pull down H3K27ac modified chromatin, then eluted and reversed cross-linking, purified and sequenced on Illumina sequencing platform. First, FastQC-v0.11.8 software was used for quality control. In brief, clean reads were aligned to the human reference genome assembly GRCh38 (hg38), and only unique mapped reads were used for subsequent analysis. The H3K27ac ChIP-seq enriched regions were then determined by model-based analysis of MACS-v1.4.2. Called peaks were filtered to exclude blacklist regions from ENCODE, the output from which the bigWig file was derived. Moreover, comparisons between paired ChIP-seq peak signals under different conditions were performed by MAnorm- 1.3.0, and the M value, *P*-value, and read density for each sample were calculated. Based on nearest gene-mapping methods, we use bedtools-v2.29.2 to assign enhancer elements

to genes. Constitutive enhancers that occur within 12.5 kilobases (kb) are further stitched together by ranking of super-enhancers for SE recognition. Stitched enhancers were assigned to the nearest genes and classified as SEs or LEs by ranking the H3K27ac signal.

## RNA sequencing and data analysis

Total RNA was extracted from NNP, DNP, and TDNP cells and intervertebral disc tissues of rats using TRIzol (Invitrogen), followed by RNA library construction and sequencing by BGISEQ-500 platform. The in-house software SOAPnuke-v1.5.2 is used to filter low-quality reads. Use Bowtie2-v2.2.5 to map clean reads to reference transcripts. Differentially expressed genes were identified by the DESeq2 algorithm. Fold change of ≤−2 or ≥2 and q value < 0.05 was regarded as significantly differentially expressed.

## MRI

MRI scans were performed 2 wk after the first operation. MRI parameter settings were determined by referring to the relevant literature.

## Live and dead cell staining

NP cells were seeded in 12-well culture plates. Pre-treatment with different concentrations of THZ1 for 2 h, the cells were stimulated with 50 ng/ml TNF-α for 24 h, and then live/dead staining was performed using the Calcein/PI Cell Viability/Cytotoxicity Assay Kit (C2015S; Beyotime). Double-fluorescent staining of NP cells with calcein-AM and PI was performed according to the manufacturer's instructions. A fluorescence microscope (Olympus IX71; Olympus) was used to observe and photograph live (green fluorescence) and dead (red fluorescence) cells.

## Alcian blue, toluidine blue, and Sirius red staining

The NP cells and rat IDD models were stained histologically with Alcian blue, toluidine blue, and Sirius red staining standard staining kit (G1560; G3668; G1472; Solarbio) in accordance with instructions of the manufacturer.

## Senescence-associated $\beta$-galactosidase(SA-$\beta$-gal) staining

$\beta$-Galactosidase staining kit (BC2585; Solarbio) was used to stain NNP, DNP, and TDNP cells in accordance with instructions of the manufacturer.

## Immunofluorescence (IF)

Human NP cells were seeded on coverslips, exposed to 50 ng/ml TNF-α, and treated with or without 50 nM THZ1, 4% paraformaldehyde was used to fix cells for 10 min, 0.1% Triton X-100 permeabilization for 10 min, 10% goat serum block for 1 h, and PBS was used to wash cells three times at intervals before incubation with MMP13 (1:3,000 dilution; ab39012; Abcam), Pol II Ser5P (1:1,000 dilution; ab5408; Abcam) at 4°C overnight. After washing three times with PBS, samples were incubated with Cy5-conjugated goat anti-

rabbit IgG or goat anti-mouse IgG (1:200; Abcam) in PBS for 1 h, and then stained with DAPI for 10 min at room temperature (Life Technologies). Fluorescence signals were imaged using a fluorescence microscope (Olympus IX71; Olympus).

## qRT-PCR

Total RNA was extracted from NP cells or the whole intervertebral disc using TRIzol (Invitrogen). After RNA isolation, the cDNA synthesis kit (Takara) is used for reverse transcription. Gene-specific primers are provided in Table S1.

## Western blotting

Nucleus pulposus cells were seeded onto six-well plates ($5 \times 10^5$ cells per well), cells were pretreated with or without 50 nM THZ1 for 2 h, before exposed to 50 ng/ml TNF-α (Peprotech) for 24 h, and untreated cells were used as controls. NNP, DNP, and TDNP cell samples were lysed in RIPA buffer containing protease inhibitor cocktail (Roche). After 30 min incubation at 4°C, electrophoretic separation, transfer, immunoblotting, and visualization of protein bands were performed as before, with primary antibodies specific for Ser5P Pol II CTD (1:1,000 dilution; ab5408; Abcam) and MMP-13 (1: 3,000 dilution; ab39012; Abcam). Secondary antibodies were goat anti-mouse IgG (H+L), HRP (1:2,000 dilution; TE262980; Invitrogen) and goat anti-rabbit IgG (H+L), HRP (1:2,000 dilution; TK274616; Invitrogen).

## Statistical analysis

GraphPad Prism software (version 7.0) and R software (version 3.6.3) were used for statistical analysis. $t$ test or one-way ANOVA was used to analyze the differences between groups. The screening criteria for statistically significant differences are $*P < 0.05$ and $**P < 0.01$, $***P < 0.001$, $****P < 0.0001$.

# Data Availability

The raw data supporting the conclusions of this article will be made available by the authors upon request.

# Supplementary Information

# Acknowledgements

This work was supported by the National Natural Science Foundation of China (82072491); National Natural Science Foundation of China (31900967); Natural Science Foundation of Tianjin city (20JCYBJC00820).

## Author Contributions

G Li: conceptualization, data curation, formal analysis, investigation, visualization, methodology, and writing—original draft, review, and editing.
Y Kang: resources and data curation.
X Feng: resources, data curation, software, validation, investigation, visualization, and methodology.
G Wang: formal analysis, investigation, visualization, and methodology.
Y Yuan: data curation, validation, and methodology.
Z Li: formal analysis, visualization, and methodology.
L Du: funding acquisition and methodology.
B Xu: conceptualization, formal analysis, supervision, funding acquisition, project administration, and writing—review and editing.

## Conflict of Interest Statement

The authors declare that they have no conflict of interest.

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
