## [Reviewer comments · Life Science Alliance]

Life Science Alliance

Dynamic changes of enhancer and super enhancer landscape in degenerated nucleus pulposus cells

Guowang Li, Yuxiang Kang, Xiangling Feng, Guohua Wang, Yue Yuan, Zhenhua Li, Lilong Du, and Baoshan Xu
DOI: <https://doi.org/10.26508/lsa.202201854>

Corresponding author(s): Baoshan Xu, Tianjin Hospital

Review Timeline:

Submission Date:	2022-11-25
Editorial Decision:	2023-02-09
Revision Received:	2023-03-07
Editorial Decision:	2023-03-21
Revision Received:	2023-03-23
Accepted:	2023-03-24

Scientific Editor: Novella Guidi

Transaction Report:

February 9, 2023

Re: Life Science Alliance manuscript #LSA-2022-01854-T

Prof. Baoshan Xu
Tianjin Hospital
406 Jiefang South Road, Hexi District, Tianjin
Tianjin 300202
China

Dear Dr. Xu,

Thank you for submitting your manuscript entitled "Dynamic changes of enhancer and super enhancer landscape in degenerated nucleus pulposus cells" to Life Science Alliance. The manuscript was assessed by expert reviewers, whose comments are appended to this letter. We invite you to submit a revised manuscript addressing the Reviewer comments.

Thank you for this interesting contribution to Life Science Alliance. We are looking forward to receiving your revised manuscript.

Sincerely,

B. MANUSCRIPT ORGANIZATION AND FORMATTING:

Reviewer #1 (Comments to the Authors (Required)):

Super-enhancers are large clusters of multiple proximal enhancers that are enriched for high densities of transcription factors, cofactors, and enhancer epigenetic modifications. Studies have found that SE-regulated transcription is dependent on the CDK7, the underlying mechanism of transcriptional abnormalities in intervertebral disc degeneration and whether CDK7 inhibitors can delay intervertebral disc degeneration remain unexplored. This research used the tools of Chromatin immunoprecipitation sequencing and RNA sequencing, and further verified them through cell and animal tests with supportive data, which provided the first characterization of enhancer and super-enhancer landscapes before and after the degeneration of NP cells and important insights into delaying IDD progression. The selected topic is close to the intervention of IDD and conforms to the contribution direction of the journal. But there are still some questions that need to be addressed.

1. The scale bar should be added for the WB results;

2. There are some unreferenced statements. Please keep the reference accordance suitable;

3. In the discussion part, the previous researches and discussions regarding the characterization of enhancer and super-enhancer landscapes are recommended to demonstrate;

The language of the full text needs to be thoroughly revised, including by native speakers or English language professionals.

Reviewer #2 (Comments to the Authors (Required)):

This article showed the dynamic changes of enhancer and super enhancer landscape in degenerated nucleus pulposus cells, and revealed that THZ1, a CDK7 inhibitor, could significantly antagonize IDD development through inhibiting the SE complex component. This article has significant scientific significance and provides a new idea for the treatment of IDD.

However, there are still two questions that bother me:

1. Why was CDK7 chosen for further study, rather than CDK9 or even BRD4?

2. It'd better to assess the histological scores of the result of Figure 7b by using the intervertebral disc scoring system.

To Reviewer #1:

Comment 1: The scale bar should be added for the WB results;

Reply 1: Thank you for your reminding. We have supplemented the quantitative analysis of WB results.

Changes in the text: Quantitative analysis has been shown on the underside of WB results in Figure 5C, and the legend in Figure 5C was also supplemented (see page 36, line 770-771).

Comment 2: There are some unreferenced statements. Please keep the reference accordance suitable;

Reply 2: Thank you very much for your reminding. We have reviewed the references one by one, and revised and reordered the references.

Changes in the text: References in the manuscript and corresponding numbers in the text have been revised (see page 31-34, line 638-718).

Comment 3: In the discussion part, the previous researches and discussions regarding the characterization of enhancer and super-enhancer landscapes are recommended to demonstrate;

Reply 3: Thank you for your reminder. In the discussion section, we have

added the relevant descriptions about the characteristics of enhancers and super enhancers.

Changes in the text: We have modified our text as advised (see page 21, line 424-440).

Comment 4: The language of the full text needs to be thoroughly revised, including by native speakers or English language professionals.

Reply 4: Thank you for your suggestion. We have polished the language.

Changes in the text: We have modified our text as advised.

To Reviewer #2:

Comment 1: Why was CDK7 chosen for further study, rather than CDK9 or even BRD4?

Reply 1: Current studies have found that SE-regulated transcription is dependent on bromodomain-containing protein 4 (BRD4), the Mediator complex, the TF IIH complex containing cyclin-dependent kinase 7 (CDK7), and the transcription elongation complex (P-TEFb) containing CDK9. CDK7 initiates transcription by phosphorylation of serine 5th of Pol II C-terminal domain (CTD); CDK9 mainly phosphorylates serine

2nd of Pol II CTD to promote transcriptionally paused Pol II to enter the transcription elongation stage, also known as Pol II release. In addition, BRD4 promotes the assembly of super-enhancers by recruiting the Mediator complex and thus promote the release of Pol II from the paused state. Therefore, it is generally believed that the key regulatory points of SE regulation of transcription, the Mediator complex, BRD4 and key CDKs, are potential to be developed as new targets for the treatment of diseases

Therefore, inhibitors related to CDK7, CDK9 and BRD4 may retard the degeneration of intervertebral disc. However, inhibitors of CDK9 and BRD4 have been studied for intervertebral disc degeneration, including the first effective and highly selective P-TEFb/CDK9 inhibitor, called atuvaciclib (BAY-1143572), can effectively reduce the inflammatory reaction in intervertebral disc degeneration through CDK9 inhibition(1); BRD4 may suppress MAPK and NF- κ B signaling and activate autophagy to suppress MMP-13 expression in diabetic intervertebral disc degeneration, and diabetic intervertebral disc degeneration may be compromised by BRD4 inhibitors(2); BRD4 inhibition reduced NP cell senescence and apoptosis by induced autophagy, which ultimately alleviated intervertebral disc degeneration(3). The role of CDK7 in intervertebral disc degeneration has not been studied, so we chose CDK7 as the target.

Changes in the text: None.

Comment 2: It'd better to assess the histological scores of the result of Figure 7b by using the intervertebral disc scoring system.

Reply 2: Thanks for the reminder, we have supplemented the quantitative analysis of histological scores to Figure 7B, and the corresponding legend of Figure 7B has also been described.

Changes in the text: Quantitative analysis has been shown in the right of Figure 7B, we have modified our text as advised (see page 37, line 795-796).

1. Ni W, Zhang F, Zheng L, Wang L, Liang Y, Ding Y, et al. Cyclin-Dependent Kinase 9 (CDK9) Inhibitor Atuveclib Suppresses Intervertebral Disk Degeneration via the Inhibition of the NF- κ B Signaling Pathway. *Frontiers in Cell and Developmental Biology*. 2020;8.
2. Wang J, Hu J, Chen X, Huang C, Lin J, Shao Z, et al. BRD4 inhibition regulates MAPK, NF- κ B signals, and autophagy to suppress MMP-13 expression in diabetic intervertebral disc degeneration. *The FASEB Journal*. 2019;33(10):11555-66.
3. Zhang G-Z, Chen H-W, Deng Y-j, Liu M-Q, Wu Z-L, Ma Z-J, et al. BRD4 Inhibition Suppresses Senescence and Apoptosis of Nucleus Pulposus Cells by Inducing Autophagy during Intervertebral Disc Degeneration: An In Vitro and In Vivo Study. *Oxidative Medicine and Cellular Longevity*. 2022;2022:9181412.

March 21, 2023

RE: Life Science Alliance Manuscript #LSA-2022-01854-TR

Prof. Baoshan Xu
Tianjin Hospital
406 Jiefang South Road, Hexi District, Tianjin
Tianjin, Tianjin 300202
China

Dear Dr. Xu,

Thank you for submitting your revised manuscript entitled "Dynamic changes of enhancer and super enhancer landscape in degenerated nucleus pulposus cells". We would be happy to publish your paper in Life Science Alliance pending final revisions necessary to meet our formatting guidelines.

- please add ORCID ID for corresponding author-you should have received instructions on how to do so
- please add a summary blurb/alternate abstract and Keywords to our manuscript system
- please add the Twitter handle of your host institute/organization as well as your own or/and one of the authors in our system
- please use the [10 author names, et al.] format in your references (i.e. limit the author names to the first 10)

A. FINAL FILES:

B. MANUSCRIPT ORGANIZATION AND FORMATTING:

Sincerely,

Reviewer #1 (Comments to the Authors (Required)):

This revised manuscript is eligible to publication.

Reviewer #2 (Comments to the Authors (Required)):

The authors answered my questions well and revised the manuscript. I recommend that the manuscript could be accepted without any revision.

March 24, 2023

RE: Life Science Alliance Manuscript #LSA-2022-01854-TRR

Prof. Baoshan Xu
Tianjin Hospital
406 Jiefang South Road, Hexi District, Tianjin
Tianjin, Tianjin 300202
China

Dear Dr. Xu,

Thank you for submitting your Research Article entitled "Dynamic changes of enhancer and super enhancer landscape in degenerated nucleus pulposus cells". It is a pleasure to let you know that your manuscript is now accepted for publication in Life Science Alliance. Congratulations on this interesting work.

DISTRIBUTION OF MATERIALS:

Again, congratulations on a very nice paper. I hope you found the review process to be constructive and are pleased with how the manuscript was handled editorially. We look forward to future exciting submissions from your lab.

Sincerely,
